# Do ADHD Symptoms, Executive Function, and Study Strategies Predict Temporal Reward Discounting in College Students with Varying Levels of ADHD Symptoms? A Pilot Study

**DOI:** 10.3390/brainsci11020181

**Published:** 2021-02-02

**Authors:** Anouk Scheres, Mary V. Solanto

**Affiliations:** 1Behavioural Science Institute, Radboud University, 6525 GA Nijmegen, The Netherlands; 2Zucker School of Medicine at Hofstra-Northwell, New York, NY 11042, USA; msolanto@northwell.edu

**Keywords:** ADHD, college students, executive function, temporal discounting, reward, delay discounting, study strategies

## Abstract

The purpose of this study was to examine the relationship between temporal reward discounting and attention deficit hyperactivity disorder (ADHD) symptoms in college students. Additionally, we examined whether temporal reward discounting was associated with executive functioning in daily life and with learning and study strategies in this group. Thirty-nine college students (19 with ADHD and 20 controls) participated after meeting criteria for ADHD or non-ADHD based on standardized assessment. Strong preferences for small immediate rewards were specifically associated with the ADHD symptom domain hyperactivity–impulsivity. Additionally, these preferences were associated with daily life executive function problems and with weak learning and study strategies. This suggests that steep temporal discounting may be a key mechanism playing a role in the daily life challenges that college students with ADHD symptoms face. If these findings are replicated in larger samples, then intervention strategies may profitably be developed to counteract this strong preference for small immediate rewards in college students with ADHD symptoms.

## 1. Introduction

Attention deficit hyperactivity disorder (ADHD) is a condition that is characterized by symptoms of inattention and/or hyperactivity–impulsivity [1]). It is viewed as a neurodevelopmental disorder that persists into adulthood in 50–70% of cases [2]. Clinically speaking, while ADHD is a diagnostic category, the expression of symptoms appears to vary along a continuum across the population and therefore it may be best viewed as a dimensional construct [3]. ADHD symptoms are associated with relative weakness in executive functions (EF) such as response inhibition, organizing and planning, and working memory (e.g., [4,5,6]).

One sub-group of individuals with ADHD symptoms for whom these deficits in EF may be of particular relevance and are relatively under-studied are college students. There has been an increase in the number of individuals with ADHD who are attending college [7]. Approximately 2–8% of college students report symptoms of ADHD that reach clinically relevant levels [8]. Although brain maturation is still ongoing during emerging adulthood, and cortical maturation is delayed in individuals with ADHD [9,10], the demand on EF is high during this phase of life. Often, college students start living independently, increasing the demand on EF. Indeed, it has been demonstrated that living at home is a protective factor for college students with ADHD with respect to impairment and college maladjustment [11]. In addition, college students start managing their finances independently, and managing, planning, and organizing their academic tasks independently [12,13,14,15], placing a demand on EF.

The body of knowledge about the functioning and manifestation of symptoms in college students and the role that EF plays in this is increasing [16]. College students with ADHD, compared to students without ADHD, have lower achievement/grade point averages (GPA) [17,18,19], and this link between ADHD and low GPA was fully mediated by weak study skills, as measured with the Learning and Study Strategies Inventory (LASSI), a self-report measure assessing strategic learning [20]. Additionally, college students with ADHD withdraw from more classes than their peers without ADHD [17]. They are generally at increased risk for academic, emotional, and psychological problems [21,22,23] and report lower quality of life [24]. Weyandt and colleagues [16] found EF deficits in college students on both a task-based measure and on self-report. Note, however, that impairment is not always evident on test measures of academic and cognitive functioning, but may be more clearly captured by self-reported measures of symptom manifestation, EF and psychological distress in daily life [21]. Associations between self-reported ADHD symptoms and EF weaknesses have been reported [25,26], and persisted after controlling for poor sleep quality [25]. Self-reported EF problems have also been shown to mediate the association between ADHD and impairment [27]. In sum, evidence is building that shows that college students with ADHD suffer from weaknesses in EF, contributing to the impairment that they experience in daily life and academic functioning.

A mechanism that may be central to the difficulties that college students with ADHD experience is steep temporal reward discounting. Temporal reward discounting (TD) is defined as the extent to which a large, delayed reward loses its perceived value as a function of the time needed to wait for it (see [28] for review). Many choices that people face in life require a trade-off between the anticipated benefits/rewards and costs of two options at different points in time. Students, for example, may frequently be faced with a choice between studying now in order to obtain good grades on a test or paper vs. pursuing appealing short-term rewards such as use of social media or partying with friends. Over many iterations, the pursuit of immediate over more significant but delayed rewards may account for the underachievement and underperformance educationally and occupationally that is well-documented in adult outcome studies of individuals with ADHD (see, for example, [29]).

TD is typically measured by asking participants to choose between a small reward that is available immediately and a larger reward that will not be available until some point in the future. The neural circuitries that are involved include the ventral striatum, medial prefrontal cortex, posterior cingulate cortex (valuation of immediate rewards) and the dorsolateral prefrontal cortex, posterior parietal cortex, and ventrolateral prefrontal cortex (valuation of both immediate and delayed rewards) [28]. Given that reduced ventral striatum responsivity has been consistently demonstrated in individuals with ADHD during reward anticipation tasks (see [30] for meta-analysis), it is possible that this brain region also plays a role in steep TD in those with ADHD (see [31]). Additionally, hyperactivation of the amygdala likely plays a role, as this has been observed in individuals with ADHD during the processing of delayed rewards [31]. Similarly, relatively strong amygdala reactivity to increasing delay durations in temporal discounting tasks was reported [32]. Finally, in a task specifically designed to assess delay aversion, adolescents with ADHD demonstrated stronger delay-related increases in amygdala activation than typically developing adolescents [33]. The relevance of the temporal discounting construct is illustrated by the demonstrated links between steep TD and conditions characterized by atypical impulse control, including substance abuse [34,35], pathological gambling [36], obesity [37], and ADHD [38,39].

Two recent meta-analyses have shown that, when choosing between larger money amounts that are available in the future and smaller money amounts that are available now, individuals with ADHD discount the larger delayed reward more strongly than individuals without ADHD, and thus demonstrate a stronger relative preference for smaller immediate rewards [38,39], supporting the dual pathway model of ADHD [40]. Patros et al. [39] limited the meta-analyses to children and adolescents, whereas Jackson and MacKillop [38] also included the more limited number of studies with adults. Both meta-analyses reported highly significant differences between groups, with medium effect sizes reflecting half a standard deviation difference between individuals with a clinical diagnosis of ADHD and controls. Additionally, effect sizes were homogenous across studies and importantly, there was no evidence of publication bias.

The findings so far appear to be very similar in adults with ADHD (relative to controls) as in children and adolescents with ADHD (relative to controls). Jackson and MacKillop [38] reported effect sizes of *d* = 0.47 for participants <18 years of age, and *d* = 0.40 for participants >18 years of age. However, college students are extremely under-represented in these studies; in six of the seven studies reviewed, the adult participants were recruited through clinics [31,41,42,43,44,45]. Only one study [46] had a specific focus on the recruitment of college students with and without ADHD. In that study, the presence of a diagnosis was subjectively reported by college students without measurement of ADHD symptom levels and/or confirmation of meeting the criteria for a diagnosis by a diagnostic assessment. If we want to understand whether steep TD is indeed a mechanism that is central to the difficulties (in EF and study skills) that college students with high levels of ADHD symptoms experience, then we need to study TD in this population. Therefore, more research on TD in college students with clinically assessed symptoms/diagnosis of ADHD is clearly needed, because the construct of TD seems to be relevant for the sort of trade-offs that college students repeatedly have to make. If indeed it turns out that steep TD is a mechanism that is associated with ADHD in college students, interventions may be developed to target this mechanism in order to help students make better trade-offs between the future and the “now” in such a way as to increase psychological wellbeing and achievement of academic and other longer-term goals.

Therefore, the first aim of this study was to examine the association between TD of rewards and ADHD symptoms in college students. In order to have a sufficient range in ADHD symptoms, college students both with and without a clinical diagnosis of ADHD, as confirmed by a standardized assessment, were included. Based on the previously reported meta-analyses [38,39] and the one study in college students in which a relatively steep TD was observed in those with a self-reported diagnosis of ADHD [46], we hypothesized that higher levels of ADHD symptoms in college students would be associated with stronger preferences for small immediate rewards. Additionally, given that steep TD is generally viewed as an operational manifestation of impulsivity [47], we expected that steep TD would be particularly associated with symptoms of hyperactivity–impulsivity and not, or to a lesser extent, with symptoms of inattention. Little is known so far about the specific association between TD and ADHD symptom domains [38].

Additionally, in order to establish whether steep TD is central to the daily life EF problems experienced by college students with high levels of ADHD symptoms, we examined whether TD and daily life EF were associated and hypothesized that daily life EF problems would correlate with steep TD. Finally, because academic functioning is a crucial outcome measure for students in general, and because it is known that students with high levels of ADHD symptoms utilize suboptimal study strategies [48], and since weak executive functioning exacerbates impairment in college students with ADHD [49], we examined whether steep TD is associated with weak learning and study strategies.

In order to pursue these aims, we administered a TD task to college students who were rigorously assessed and diagnosed with ADHD according to a standardized procedure, and who subsequently participated in an intervention study [50]. We also administered this TD task to a convenience sample of students without ADHD on the same campus. In addition to the diagnostic instruments regarding inattention and hyperactivity–impulsivity, we administered self-report questionnaires to assess daily life EF, as well as learning and study strategies. Associations between ADHD symptoms, daily life EF, and learning/study strategies on the one hand, and TD on the other, were examined.

## 2. Materials and Methods

### Participants

Participants were 39 college students (Bachelors or Masters level) from Dutch universities, 19 with a diagnosis of ADHD and 20 typically developing students without ADHD. Participants were required to be between the ages of 18 and 30, and able to understand instructions in English. Twenty-nine participants were female, and ten were male. The mean age was 21.40 (standard deviation (SD): 2.62). The participants with ADHD (16% Masters students and 84% Bachelors students) were recruited through the Universities’ special needs departments and the Department of Psychiatry at the University Medical Center for an intervention study that entailed 12 weeks of group cognitive behavioral therapy [50], and were rigorously diagnosed with diagnostic interviews and rating scales (for details, see [50]). The typically developing students (100% Bachelors students) were recruited through a university experiment participation system which is primarily aimed at Bachelors students enrolled in courses in psychology and pedagogy. All typically developing students were first-year students enrolled in a Bachelors program of psychology. Questionnaires were used to determine that these students had scores within the normal range (less than 1.0 SD above the mean) on ADHD questionnaires. Unintentionally, the typically developing participants were younger (mean (M) = 19.85, SD = 1.31) than the participants with ADHD (M = 23.74; SD = 2.73), and as a result, age was confounded with ADHD symptoms. Similarly, the typically developing participants (from psychology and pedagogy) had a higher proportion of females (95%) than the participants with ADHD (53%) who came from all study programs. Here, we report on the baseline assessment of TD in the ADHD group and a typically developing group without ADHD, treating these 39 participants as one sample.

## 3. Materials

### 3.1. Temporal Discounting Task

A hypothetical TD task was administered on which participants made repeated choices between hypothetical small, variable money amounts (€5, €10, €20, €30, €40, €50, €60, €70, €80, €90, €95) that would be delivered today, and a hypothetical large constant money amount that would be delivered after 1 year (€100). For example, on some trials, participants chose between the hypothetical options of receiving €50 today or €100 after 1 year. Each hypothetical small immediate reward was paired twice with the large, delayed reward, resulting in 22 choice trials. Trials were presented in the same pseudo-random order to all participants. Choices were short sentences presented on a computer screen (e.g., “€50 today” or “€100 after 1 year”), with one option displayed to the left and the other option presented to the right of a fixation cross. The left or right position of the delayed option was balanced over the trials. Participants made a choice by pressing the button corresponding to the preferred option (pressing (with the left index finger) the ”A” key on the keyboard for the option on the left, and pressing (with the right index finger) the “L” key on the keyboard for the option on the right).

Participants were informed that all trials were hypothetical, which meant that they would not actually have to wait and they would not actually receive the money amounts. Participants were asked to choose the way they would if these choices were real. The dependent variable was the subjective value of €100 after 1 year, as determined based on the switch point that was observed in the choice pattern. For example, if a participant preferred to receive €100 after 1 year over €5, €10, €20, €30, €40, and €50 now, but this preference for the delayed reward switched to preferring the immediate reward when its amount was €60, €70, €80, €90, or €95, then the subjective value of €100 after 1 year was determined to be €55 (i.e., the mean of €50 and €60 (see [51])).

### 3.2. Conners Adult ADHD Rating Scale–Long Version Self-Report (CAARS)

The Conners Adult ADHD Rating Scale – Long Version Self-Report (CAARS) [52] was used to assess symptoms of inattention and symptoms of hyperactivity–impulsivity. The DSM-scale for inattention and the DSM-scale for hyperactivity–impulsivity were used. Raw scores were converted to T-scores, based on age- and gender-based norms.

### 3.3. Daily Life Executive Functions: Barkley Deficits in Executive Functioning Scale (BDEFS)

The Barkley Deficits in Executive Functioning Scale (BDEFS) for adults [53] is an 89-item assessment tool in which participants can report the extent to which they experienced trouble with executive functioning in their daily lives in the past 6 months. Executive functioning difficulties are listed, and participants can indicate how often they experience these on a 4-point scale (never or rarely; sometimes; often; very often). Five dimensions are measured, namely Self-Management to Time (21 items; example item: “waste or mismanage my time”), Self-Organization/Problem Solving (24 items; example items: “I have trouble organizing my thoughts”, “I am slower than others at solving problems I encounter in my daily life”), Self-Restraint (18 items, example item: “Unable to inhibit my reactions or responses to events or others”), Self-Motivation (12 items, example item: “Others tell me I am lazy or unmotivated”), and Self-Regulation of Emotions (12 items, example item: “Quick to get angry or become upset”). For each dimension, a total score can be computed and, based on gender- and age-specific norms, percentile scores can be derived. Higher percentile scores indicate greater difficulty. This instrument has good internal consistency and test–retest reliability, and discriminates considerably between adults with ADHD and adults in the general population [53]. It offers high ecological validity, given its focus on daily functioning, and as illustrated by its predictive power of impairments in major life activities. For the aim of the current study, the percentile scores for the total EF summary score were used (total of the five dimensions), with higher scores reflecting larger difficulties. In addition, the scores for each of the five dimensions were entered into secondary analyses.

### 3.4. Learning: The Learning and Study Strategies Inventory (LASSI)

The LASSI [54] is a 60-item survey that assesses students’ awareness of the use of learning and study strategies. This awareness and these learning strategies are related to three components: (1) Skill, (2) Will, and (3) Self-Regulation. The survey consists of 10 scales, represented by 6 items each. The items focus on covert and overt thoughts, behaviors, attitudes, motivations, and beliefs that are related to successful learning. Students can indicate for each item on a 5-point scale the extent to which a certain strategy is typical for them (ranging from “not at all typical of me” to “very much typical of me”). Based on norms, scale scores are converted to percentile scores, with higher percentiles indicating higher Skill/Will/Self-Regulation. The internal consistency of each of the LASSI scales is good, with Cronbach’s alpha values ranging from 0.76–0.87.

The scales related to the Skill component are: Information Processing (e.g., “I try to find relationships between what I am learning and what I already know”), Selecting Main Ideas (e.g., “I have difficulty identifying the important points in my reading”), and Test Strategies (e.g., “I have difficulty adapting my studying to different types of courses”). The percentile scores for each of these 3 scales were averaged to form a Skill percentile score. The scales related to the Will component are: Anxiety (e.g., “I feel very panicky when I take an important test”), Attitude (e.g., “I only study the subjects I like”), and Motivation (e.g., “When work is difficult I either give up or study only the easy parts”). The percentile scores for each of these 3 scales were averaged to form a Will percentile score. The scales related to the Self-Regulation component are: Concentration (e.g., “My mind wanders a lot when I study”), Self-Testing (e.g., “I stop periodically while reading and mentally go over or review what was said”), Time Management (e.g., “I find it hard to stick to a study schedule”), and Using Academic Resources (e.g., “I am not comfortable asking for help from instructors in my courses”). The percentile scores for each of these 4 scales were averaged to form a Self-Regulation percentile score. For the primary analysis, the LASSI total score (across components) was used, with lower scores reflecting greater difficulties. For the secondary analyses, the scores for LASSI Skill, LASSI Will, and LASSI Self-Regulation were examined separately.

## 4. Procedure

This study was conducted in accordance with the ethical guidelines that are outlined in the Declaration of Helsinki. The Ethics Committee of the Radboud University Faculty of Social Sciences approved this study (approval code: ECSW2017-0805-51). All participants were given comprehensive information about the study beforehand and provided active written consent.

## 5. Analyses

### 5.1. Descriptive Statistics

Means and standard deviations for the variables of interest were computed.

### 5.2. Correlations

Pearson correlations between the variables of interest were computed. The correlations between age and the variables of interest were also computed.

### 5.3. Primary Analyses

Regression analysis was used to examine the predictive value of the ADHD symptom domains (inattention and hyperactivity–impulsivity) to TD. In the first regression, the subjective value of €100 after 1 year was entered as the dependent variable, and inattention (Conners Adult ADHD Rating Scale Diagnostic and Statistical Manual 5th edition (CAARS DSM-V) Inattention symptom scale T-score) was entered as the predictor. In the second regression, hyperactivity–impulsivity (CAARS DSM-V hyperactive–impulsive symptom scale) was entered as the predictor.

In order to examine the extent to which daily life EF predicted TD, the subjective value of €100 after 1 year was entered as the dependent variable, and the percentile score of total EF was entered as the predictor. To study the relationship between TD and learning and study strategies, regression analysis was used with subjective value of €100 after 1 year as the dependent variable, and the LASSI total score as predictor.

In order to determine whether EF total and/or LASSI total explained additional variance in TD over and above the variance accounted for by ADHD hyperactivity–impulsivity, hierarchical regression was performed in which ADHD hyperactivity–impulsivity, the prior predictor, was entered first, followed by the other variables.

Because of the high overlap between age/gender and the predictors of interest (collinearity), and because age/gender were not associated with the dependent variable, these were not included in the regression models.

### 5.4. Secondary Analyses

In order to gain insight into which subscales of the BDEFS and the LASSI were associated with TD, secondary stepwise regression analyses were conducted: one in which the subscales of the BDEFS were entered as predictors, and one in which the subscales of the LASSI were entered as predictors. The subjective value of €100 after 1 year was entered as the dependent variable. The criterion for entry was *p* < 0.05, and the criterion for removal was *p* > 0.10.

## 6. Results

### 6.1. Descriptive Statistics

Means and standard deviations of the variables of interest are reported in Table 1.

### 6.2. Correlations

Correlations between the variables are reported in Table 2.

### 6.3. Primary Analyses

Inattention was weakly and negatively associated with subjective value, with inattention accounting for 5% of the variance in discounting, but the regression function did not reach statistical significance (Table 3, Figure 1).

Hyperactivity–impulsivity was negatively and moderately associated with subjective value, with hyperactivity–impulsivity accounting for 10% of the variance in discounting, and the regression function (*R*^2^) reached statistical significance. Participants’ subjective value of €100 after 1 year decreased by €0.42 (unstandardized *B*) for each point the hyperactivity–impulsivity score went up. See Table 3 and Figure 1 for details.

EF total significantly predicted discounting, with EF problems being associated with a lower subjective value of €100 after 1 year (*R*^2^ = 0.14; see Table 3). Participants’ subjective value of €100 after 1 year decreased by €0.26 (unstandardized B) for each point of increase in the BDEFS total (percentile) score.

Learning and study strategies significantly predicted TD, explaining 10% of the variance (Table 3). Participants’ subjective value of €100 after 1 year decreased by €0.27 (unstandardized B) for each point of decrease in the LASSI score.

In the hierarchical regression, neither the BDEFS total nor the LASSI total explained significant additional variance in TD over and above the variance already accounted for by ADHD Hyperactivity–Impulsivity, which was entered at step 1. Results for the BDEFS entered at step 2 were as follows: ΔF (1.36) = 1.85, ΔR^2^ = 0.04, *p* = 0.18; and when entered at step 3 after the LASSI: ΔF(1.35) = 0.40, ΔR^2^ = 0.01, *p* = 0.53. Parallel results for the LASSI entered at step 2 were: ΔF(1.36) = 1.45, ΔR^2^ = 0.04, *p* = 0.24; and when entered at step 3 after the BDEFS: ΔF (1.35) = 0.03, ΔR^2^ = 0.001, *p* = 0.87).

### 6.4. Secondary Analyses

#### 6.4.1. Daily Life Executive Function Subscales

Self-Motivation was the strongest daily life EF predictor of temporal discounting, as this was the only predictor that remained in the stepwise regression. It accounts for 15% of the variance in discounting (*R*^2^ = 0.15, *p* = 0.009).

#### 6.4.2. Learning/Study Strategies Subscales

The LASSI subscale Skill was the strongest predictor of TD, as this was the only predictor that remained in the stepwise regression. It accounts for 11% of its variance (*p* = 0.04).

## 7. Discussion

This pilot study examined temporal discounting in college students with varying levels of ADHD symptoms and hypothesized that hyperactivity–impulsivity would be associated with steeper TD and that this would not, or less so, be the case for inattention. Secondly, we examined whether steep TD was associated with self-reported executive functioning in daily life. Finally, we addressed the possibility that steep TD may be associated with weaknesses in learning and study strategies.

As predicted, hyperactivity–impulsivity and not inattention was significantly associated with steeper discounting. This finding supports theories proposing that this construct may be associated specifically with the Combined type of ADHD, or the symptom dimension hyperactivity–impulsivity [40,55], as well as with theoretical work that operationalizes impulsivity as steep TD [47]. This finding in college students with varying levels of hyperactivity–impulsivity symptoms extends a previous finding in children and adolescents with a clinical diagnosis of ADHD, in whom only the symptom domain hyperactivity–impulsivity was associated with steep TD [56]. These authors suggested that there is a dimensional relationship between hyperactivity–impulsivity and TD, transcending the artificial boundaries of diagnostic categories. The current finding in college students is also in line with a recent large-scale adult population study that showed a stronger link between hyperactivity and TD than inattention and TD [57]. This finding has implications for interventions that may be offered to college students with a diagnosis of ADHD: targeting the mechanism of temporal discounting may primarily be relevant for those who have high levels of symptoms of hyperactivity–impulsivity, and not for those with high levels of inattention only.

In addition to the link between hyperactivity–impulsivity and temporal discounting, we found that daily life EF was associated with the ability/willingness to wait for a larger delayed reward. This suggests that TD may be a key mechanism playing a role in the daily life executive functions that are under high demand during the college years. Notably, when exploring this association for the subscales of the BDEFS, steep TD was associated most strongly with self-motivation. Therefore, TD may be an important target for executive function interventions for college students, with and without ADHD, and most relevant in the area of motivation.

More specifically, these results point to the potential utility of interventions designed to increase the power of distant rewards to activate and sustain motivation. A current evidence-based CBT intervention for adults with ADHD [58,59] includes a module in which participants learn to select and focus on a personally meaningful long-term goal and its associated rewards, while overriding immediately available reinforcers. Whereas this strategy is just one of several included within that treatment program, it is possible to envision an intervention built entirely around this strategy in which college students practice this skill over months, as they pursue a long-term goal with the support of the therapist and group members. Further research would be necessary to confirm the success and long-term efficacy of such interventions. Inspiration could be sought from future-orientation trainings that have been developed and evaluated on their effectiveness in reducing temporal discounting (see review by Scholten and colleagues [60]). Specifically, Episodic Future Thinking training, during which participants learn to vividly imagine obtaining certain reinforcers in the distant future, appeared to be the most promising training that resulted in robust reductions in temporal discounting across a substantial number of studies. In addition, some of these effects generalized to clinical outcome measures such as smoking behavior. It will be interesting to study the effects of such training on temporal discounting and study and learning skills in college students with varying levels of ADHD symptoms.

Finally, we found that learning and study strategies were associated with temporal discounting, as steeper TD was associated with weaker learning and study strategies. This suggests that the mechanism of TD does indeed play a role in study strategies and may help us to understand the observed difficulties that students with high levels of ADHD symptoms experience with these strategies. It is noteworthy that there was room for substantial improvement in learning and study strategies, pointing to the severity of difficulties that students, especially those with high levels of ADHD symptoms, experience in the domain of learning and studying. The link between TD and learning/study skills was strongest for the Skill component. This was surprising given the fact that steep discounting may be viewed as a motivational operationalization of impulsivity, referred to as “impulsive choice” or “waiting impulsivity” [61], and therefore, we would have expected the highest correlations to be seen between discounting and the Will component, which includes items related to motivation.

A number of limitations need to be considered. First, this pilot study enrolled a small sample. These findings need to be replicated in larger samples before drawing definitive conclusions. Secondly, the control group was recruited through a system in which first and second year Bachelors students of psychology/pedagogy take part in research studies. As a result, the participants without ADHD were younger than the participants with ADHD. Similarly, there was a higher proportion of female participants among those without ADHD than those with ADHD. Therefore, age and gender are factors that are confounded with the main predictor, ADHD symptoms. Age and gender, however, did not correlate with the dependent variable TD. Adding these factors to the models would result in extremely unstable parameter estimates due to collinearity, and we opted for the typical remedy in such situations of including only the predictors of primary interest (CAARS, BDEFS, LASSI), and not the confounding variable [62,63,64,65]. Therefore, it is important to be aware that the effects reported here may be influenced by the overlap between age and ADHD hyperactivity–impulsivity, and/or gender and ADHD hyperactivity–impulsivity. Thirdly, the primary predictors ADHD symptoms, executive function problems, and study/learning strategies, overlapped substantially with one another. To illustrate, daily life executive function problems and learning and study skills did not add explained variance to the variance already explained by hyperactivity–impulsivity. Also, the high overlap means that the associations that we found between ADHD symptoms and discounting could partly be attributed to the association between executive function problems and problems in study/learning strategies on the one hand and discounting on the other. Thus, ADHD symptoms, executive function and learning/studying skills are highly inter-related, and one may wonder how useful it is to treat these separately; perhaps these are best viewed as one domain. However, examining results for self-reported learning and study skills, separately from daily EF, may be helpful in identifying the particular skills and strategies to target in the treatment of college students. Fourthly, the discounting task that was used here was hypothetical. For the current age group, the monetary amounts that seem relevant are too large to actually be paid to participants, resulting in the need for hypothetical tasks. Future research may consider combining hypothetical and real or potentially real tasks (see [28] for an overview of task formats) in order to have a broader assessment of TD. In addition, it remains unclear in the present data whether participants chose what they actually preferred, or what they thought they should prefer. In order to more accurately measure real preferences, virtual reality may be used in the future to make tasks more realistic and vivid.

## 8. Conclusions

In sum, this pilot study demonstrated that steep temporal discounting is an important construct in relation to symptoms of hyperactivity–impulsivity in college students with ADHD and controls. This finding warrants further research into the specificity of this construct for this ADHD symptom domain. In addition, both daily life executive functioning and learning and study strategies were associated with temporal discounting. This suggests that steep temporal discounting plays a role in the daily life challenges that college students face in terms of studying (especially skill-related) and daily life executive function (especially self-motivation). If the findings of this pilot study are replicated in larger samples, then intervention strategies may profitably be developed to counteract this strong preference for small immediate rewards in college students with high levels of ADHD hyperactive-impulsive symptoms. This could potentially have beneficial effects on their daily life executive functioning, as well as on their learning and study strategies.

## Figures and Tables

**Figure 1 brainsci-11-00181-f001:**
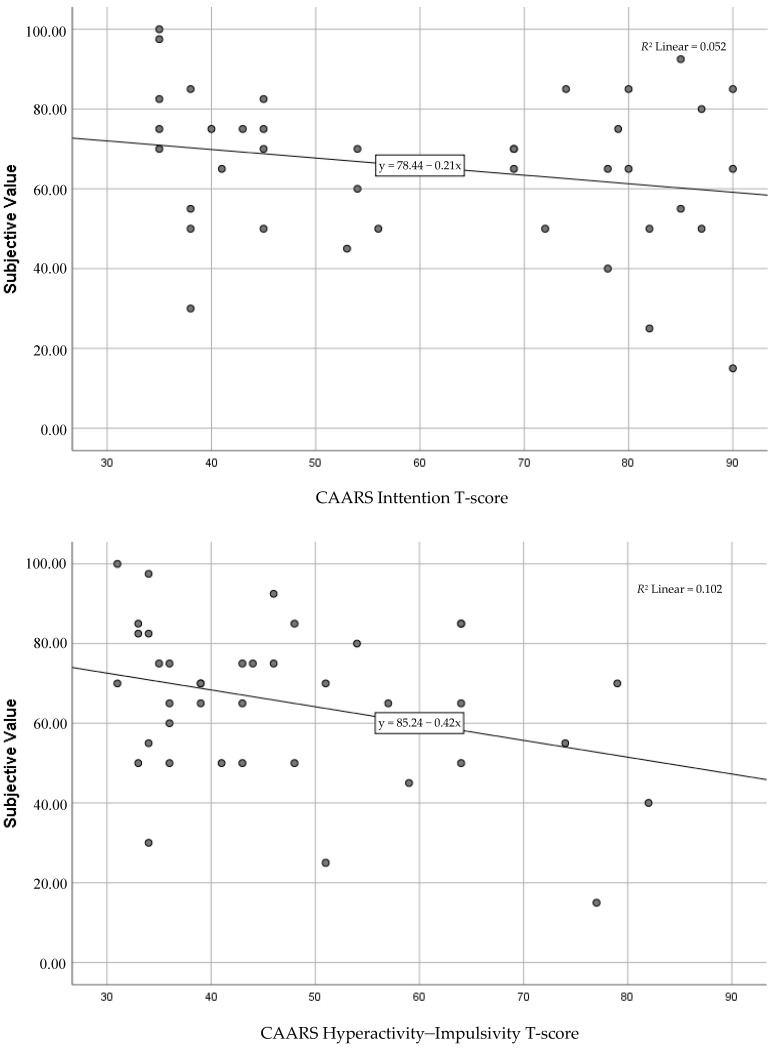
Associations between Inattention and the subjective value of €100 after 1 year (**top panel**) and Hyperactivity–Impulsivity and subje ctive value of €100 after 1 year (**bottom panel**).

**Table 1 brainsci-11-00181-t001:** Descriptors for age, attention deficit hyperactivity disorder (ADHD) symptoms, learning/study strategies, and daily life executive function (EF) problems.

	M	SD	*n* (10 M/29 F)
Age	21.40	2.62	39
CAARS			
DSM Inattention	60.87	20.45	39
DSM Hyperactive–Impulsive	47.05	14.56	39
LASSI			
Total	42.31	22.65	39
Will	46.32	25.19	39
Skill	49.71	23.52	39
Self-Regulation	33.76	25.24	39
BDEFS			
EF Total problems	70.77	27.52	39
Self-Management to Time	74.59	27.29	39
Organization	72.08	24.45	39
Self-Restraint	60.54	29.14	39
Motivation	75.33	25.71	39
Self-Regulation of Emotions	65.67	22.89	39

CAARS = Conners Adult ADHD Rating Scale-Self. Higher scores (T-scores) indicate more inattention or hyperactivity–impulsivity; LASSI = Learning and Study Strategies Inventory. Higher scores (percentile scores) indicate better skills; BDEFS = Barkley Deficits in Executive Functioning Scale. Higher scores (percentile scores) indicate more problems. M = mean. SD = standard deviation.

**Table 2 brainsci-11-00181-t002:** Correlations.

	1.	2.	3.	4.	5.	6.
1. Subjective value	1					
2. Age	−0.18	1				
3. CAARS Inattention	−0.23	0.69 **	1			
4. CAARS Hyperactivity–Impulsivity	−0.32 *	0.56 **	0.65 **	1		
5. BDEFS Total	−0.37 *	0.61 **	0.89 **	069 **	1	
6. LASSI Total	0.32 *	0.62 **	−0.85 **	−0.49 **	−0.86 **	1

* *p* < 0.05; ** *p* < 0.01.

**Table 3 brainsci-11-00181-t003:** Regression analyses with Inattention, Hyperactivity–Impulsivity, BDEFS EF total, and LASSI total as predictors and the subjective value of €100 after 1 year as the dependent variable.

	Hyperactivity–Impulsivity (*n* = 39)	Inattention (*n* = 39)	BDEFS EF Total (*n* = 39)	LASSI Total (*n* = 39)
F	4.22	2.04	5.94	4.18
β	*−0.32*	−0.23	*−0.37*	*0.32*
*R* ^2^	*0.10*	0.05	*0.14*	*0.10*
*p*	*0.047*	0.16	*0.02*	*0.05*

Meaningful associations (proportion explained variance ≥ 0.10 and *p* ≤ 0.05) in italics.

## Data Availability

The data presented in this study are available on request from the corresponding author.

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
