# Peer review of "Do ADHD Symptoms, Executive Function, and Study Strategies Predict Temporal Reward Discounting in College Students with Varying Levels of ADHD Symptoms? A Pilot Study"

_brainsci, 2021, doi:10.3390/brainsci11020181_

Round 1
Reviewer 1 Report
This is a case control study comparing ADHD and non-ADHD college students on a temporal reward discount difference paradigm.
There are major methodological problems with this paper
1) Table 1 is nowhere to be seen
2) Subjects are apparently recruited from different courses with different ages and from different years, controls seem to be a convenience sample only from first year psychology course. This may be convenient but certainly a possible bias which should be checked
3) line 131 "typical students with ADHD". Define typical
4) lines 315-317. You did not find statistically significan differences. This is all you can say and it is a basic mistake to add that this is due to the small sample size. It is up to the researcher to run a power calculation in planning the sample size before the experiment and the temporal reward discounting paradigm is well known and easily allows to to that. So one of the two consequences: a) either you did run a a power calculation and your sample is big enough and you should therefore comment on the fact that your research could not find differences in the two samples or b) you did not and therefore your results are not publishable really because of very poor methodology
Reviewer 2 Report
This is an interesting and one of the relatively few studies regarding temporal reward discounting in college students with ADHD. The authors investigated relationships with symptom domains, executive functions, and study strategies.
It's appreciable that the authors already thought and mentioned significant weak points of the study at the end of the discussion, however, few other points need to be addressed-
1- The objectives of the study need to be mentioned more clearly along with the hypothesis. 'What is not known' should be emphasized.
2- The possible gender and age effect of the study participants on the outcomes should be discussed. As per table 1, controls were almost all females (1M 19F) while both ADHD groups have a better M/F ratio. Similarly, control participants were of a lower age group with minimal spread (19.85, SD=1.31) compared to both ADHD groups (23.55, SD3.14, and 24.00, SD=2.20).
3- In figure 1, there are no Y-axis details and the p-value is also not mentioned. Although in the text, it's mentioned that the difference is not significant but, because other parameters show meaningful associations, it would be valuable to see the significance levels in subjective value.
4. Based on the limited number of participants and methodological limitations as already mentioned by authors, this study may be titled a pilot study.
Reviewer 3 Report
The paper at hand investigates temporal discounting (TD) in college students with and without ADHD, and finds that hyperactivity-impulsivity (HI, self-report) and deficits in daily life executive functioning (EF, self-report) were associated with steeper temporal discounting, while better learning and study strategies predicted less steep temporal discounting.
The introduction establishes that the study endeavor is worthwhile, as college students seem not to have been included in the samples that investigated temporal discouting in relation to adhd.
However, the sample is very, very small. Particularly the ‚experimental‘ group of people suffering from a hyperactivity-impulsive type of adhd consists of merely 8 participants, but even all adhd diagnosed students were no more than 19. I need not say more, to be honest. There is no use in interpreting group differences here. Thus, if the editor decides to go for revise and resubmit, I would suggest to cut out the lengthy reports in text and tables of 3 groups that are compared, and focus instead on the interrelations between the variables reported, possibly going into more details of associations between subscales, too. If the sample was bigger, structural equation modelling might be interesting to investigate also paths between TD, HI, and EF, and maybe even compare their relations between people with and without adhd (mainly impulsive type). But since the sample is small, I am not convinced whether the study as it is holds merit for the scientific community. Maybe computing the power of the analyses used here might reveal with how little chance the effects found can be expected to be found in the population. Also, it may not be targeted well to the readers of brain sciences, particularly given that all scales (that work) consist of self-report. Maybe a journal tailored to public health like IJERPH, or more specifically tied to adhd or college students would be more suitable.
Thus, I suggest either to publish the intercorrelations between TP, EF and HI at baseline within the scope of a larger article comparing the pre- and post-intervention scores, as the within-comparison will boost power and partially compensate for the small sample.
An alternative could be to recruit more people, which could be a convenience sample, just to explore the associations between TP, EF and HI with more power. If both more adhd patients and more ‚controls‘ are recruited, I would like to see efforts made to limit possible systematic confounds like socio-economic status or iq (though this may be taking it too far for a study as simple as this, so I will not consider it necessary).
The last, and probably most feasible, option would be to just report the intercorrelations between TP, EF, and HI across the whole sample (no tiny group comparisons), maybe also investigating the role of subscales or do a factor analysis on all these items assessing the constructs of TP, EF, and HI, which share so much conceptual overlap.
I do like the conclusions in 291-294, 346-350, and 355-359, these are honest observations and not spectacular, but worthwhile findings that may prove informative to the community. If the paper was to be published without adding anything else, I would strongly suggest to focus on the findings and build the manuscript around them. Given that the associations were found only for the full sample, the discussion should focus on how to target and improve TD, EF, and LASSI in the general population of college students with or without adhd, and please elaborate more on how the proposed interventions might look like.
Minor Issues
- line 32: Unifying Theory is not explained and only stated once, but prominently in the front as if it was explaining big time the phenomenon under question - please elaborate or delete
- line 47: the link between adhd and low gpa was mediated by weak study skills - partially or fully?
- please give sample items for scales
- p. 4, l. 147 „qualified based on the phone screen“ may be misleading, so you might better call this based on the phone screening or the like
- line 269: why use percentile groups rather than the raw score of EF as a continuous variable?
- line 294: maybe illustrating the relationship between EF and TP (and also LASSI?) with a plot or several plots would help the reader extract more of the relevant information
Round 2
Reviewer 1 Report
I find that the revised version of the manuscript satisfactorily adresses the previously highlighted issues
Reviewer 2 Report
The authors have addressed most of the concerns adequately and the article is significantly improved. No further concerns.
Reviewer 3 Report
The authors have substantially improved the manuscript, which now deems me of interest to any reader of this journal. I recognize its improvement in the present form and applaud the authors on their constructive adaptation of the manuscript.